# Nanometric Hydroxyapatite Particles as Active Ingredient for Bioinks: A Review

**Edilberto Ojeda, África García-Barrientos, Nagore Martínez de Cestafe, José María Alonso** 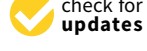**, Raúl Pérez-González and Virginia Sáez-Martínez \***

i+Med S. Coop., 01510 Vitoria-Gasteiz, Spain; eojeda@imasmed.com (E.O.); agarcia@imasmed.com (Á.G.-B.);
nmartinez@imasmed.com (N.M.d.C.); jalonso@imasmed.com (J.M.A.); rperez@imasmed.com (R.P.-G.)
**\*** Correspondence: vsaez@imasmed.com

**Abstract:** Additive manufacturing (AM), frequently cited as three-dimensional (3D) printing, is a relatively new manufacturing technique for biofabrication, also called 3D manufacture with biomaterials and cells. Recent advances in this field will facilitate further improvement of personalized healthcare solutions. In this regard, tailoring several healthcare products such as implants, prosthetics, and in vitro models, would have been extraordinarily arduous beyond these technologies. Three-dimensional-printed structures with a multiscale porosity are very interesting manufacturing processes in order to boost the capability of composite scaffolds to generate bone tissue. The use of biomimetic hydroxyapatite as the main active ingredient for bioinks is a helpful approach to obtain these advanced materials. Thus, 3D-printed biomimetic composite designs may produce supplementary biological and physical benefits. Three-dimensional bioprinting may turn to be a bright solution for regeneration of bone tissue as it enables a proper spatio-temporal organization of cells in scaffolds. Different types of bioprinting technologies and essential parameters which rule the applicability of bioinks are discussed in this review. Special focus is made on hydroxyapatite as an active ingredient for bioinks design. The goal of such bioinks is to reduce the constraints of commonly applied treatments by enhancing osteoinduction and osteoconduction, which seems to be exceptionally promising for bone regeneration.

**Keywords:** additive manufacturing; 3D printing; bioprinting; bone cements; hydroxyapatite; biomimetic

## 1. Introduction

All along the processes of biomineralization, the organism possesses the ability to produce and deposit different types of minerals with the purpose of hardening or stiffening existing tissues. Among them, calcium phosphates (CaP) can be found [1]. These salts are the dominant mineral components in vertebrate organisms in bone and tooth as well as in pathological calcifications of tissues: calculus and stones in the oral cavity and urinary system, or artherosclerotic damages.

Calcified tissues, such as bones, may be designed as anisotropic composites of natural origin. These tissues are composed of biominerals included in a protein matrix together with water and other organic materials. The mineral phase of the bone is based on several forms of calcium phosphates and constitutes 65–70% of it, water accounts for 5–8% and the organic phase, which is mainly collagen, comprises the rest [2].

The main mineral form of the mammals' bones is named biological apatite. This material is an apatite that displays a Ca/P ratio below 1.67 together with a lower amount of hydroxy groups and a higher quantity of carbonates [1]. Synthetic hydroxyapatite ($Ca_{10}(PO_4)_6(OH)_2$ or HA) is aimed to possess similarity to this biological apatite [3].

Therefore, synthetic hydroxyapatite has been commonly employed as a biomaterial for orthopaedic [4] and dental purposes as well as for improving or exchanging hard

tissues [5]. Noteworthy this material has been also tested as a drug delivery system as it displays enhanced stability and ability to target biological systems. In fact, these types of inorganic materials rely on their appropriate pore size to store a high amount of therapeutic molecules in the pores [6].

Biomaterials of synthetic origin composed of HA have been broadly researched as components of artificial bone grafts as well as surface coating agents. These materials are bioactive and biocompatible by definition and display mechanical properties and a porous structure that enables their implantation into the human body [7]. In this regard, the mechanical strength and the structure of the HA directly affects the osteointegrative, osteoinductive and osteoconductive characteristics of the material [8,9]. For example, small-sized HA crystals are rapidly dissolved, which is of great aid for skeletal disorders such as osteoporosis and other metabolic conditions [7]. Nanosized HA also receives the name HA nanoparticles and possesses a grain size of less than 100 nm in at least one dimension. HA nanoparticles display a high surface area and a fine nanostructure, close to that of the mineral discovered in hard tissues [10]. Indeed, the bioceramics which rapidly stimulate osteointegration and bone tissue generation are those that imitate the composition and structure of the bone mineral. In this regard, it has been demonstrated that ceramic biomaterials produced from HA nanoparticles show an improved resorbability and enhanced bioactivity than ceramics of micrometre range size [11,12]. The liberation of calcium ions from HA nanoparticles is comparable to that from apatite of biological origin and integrates more rapidly with tissues than that from rough crystals. In addition, some research works indicate that nanoscale HA has the potential ability to diminish apoptotic cell death and therefore to enhance the proliferation of cells and their activity attributed to the growth of bone tissue [13,14]. The enhanced proliferation and differentiation of cells may be caused by the larger surface area and the improved surface properties of HA nanoparticles compared to micron-sized ones. As a consequence cell adhesion and cell–matrix interactions are by far greater [15]. Consequently, bioceramics and biocomposites, produced from HA nanoparticles, have demonstrated to be one of the most interesting materials for a wide range of applications in biomedicine [2].

The printing of biomaterials in a controlled and precise manner is essential for the fabrication of two-dimensional (2D) and three-dimensional (3D) cell structures where biomaterials must be located between cells to hold the gravitational force. The design of the appropriate bioink is the main challenge for bioprinting. In this regard, convenient bioinks for producing active bone substituents need to display features such as biomimicry, biocompatibility, bioprintability, biodegradability and mechanical integrity [16,17]. Besides, bioprinting parameters including the effects of pressure, temperature, nozzle size of the bioprinter, bioink viscosity, the macrostructure of the resulting material (i.e., porosity) and crosslinking methods are issues of consideration for the production of substitutes of bone tissue [18,19].

## 2. Bioprinting: Technology and Suitable Biomaterials

Three-dimensional bioprinting is a type of additive manufacturing that uses biomaterials to generate 3D objects that will have a biological impact [20]. The route to 3D bioprint an item consists of several elements: data acquisition to obtain the 3D models, the material selection to bioprint and functionalization by adding other components to the item [21]. Since 1984 with the invention of stereolithography (SLA) to print objects in 3D (dimensions) to 1988 with the first bioprinting by inkjet printer to place cells by the technique of cytoscribing [22], the 3D and 3D bioprinting technology has been greatly improved. In 2002, the first bioprinting technique based on extrusion was disclosed and then commercialized as the 3D-Bioplotter [23]. From there many bioprinters and bioprinting products have been introduced, such as Tissue Scribe by 3D Cultures, LulzBot Bio by Lulzbot, BioX by Cellink, etc. Although progress has been made in these types of products, to date, it is unfeasible to obtain the 3D bioprinting of organs that possess full functionality.

Within bioprinting, there are two types of printing objects: those that involve the fabrication of non-living objects, such a prosthesis, scaffolds, medical guides, etc. [24,25] and the fabrication of living cellular objects such as cartilage, skin, nerves or bones, among other examples [20,26], where bone appears to be in a more advanced stage than other tissues for application in humans [27–29]. Three-dimensional bioprinting essentially relies on three paths: (a) extrusion, (b) droplet-mediated and (c) UV/photocuring-mediated bioprinting. A summary is displayed in Figure 1.

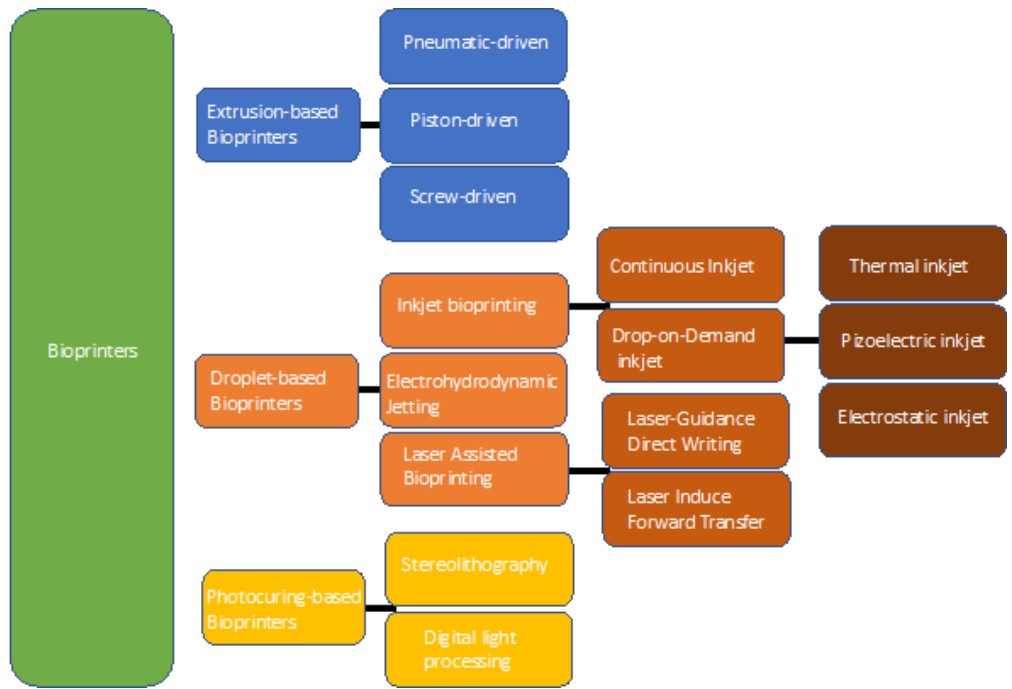

**Figure 1.** Overview of 3D bioprinting techniques.

### 2.1. Extrusion-Based Bioprinting

Bioprinting based on extrusion moves the bioink through a nozzle mechanically or pneumatically, to generate microfilaments in a continuous way that are placed on a substrate (solid or liquid) to form desired structures. This way may be employed to print a variety of biomaterials with different viscosities and concentrations of cells. There are several features, such as the diameter of the nozzle, temperature, speed of the movement, pressure and speed of the extrusion, and path interval which must be taken into account when extrusion bioprinting is employed. Depending on the extrusion method this type of bioprinting can be classified as pneumatic guided, piston guided and screw driven [30]. Extrusion, which is pneumatically guided, releases liquid through the dispensing nozzle by employing compressed air [21]. Extrusion driven by a piston is a system linked to a motor by means of a guide screw that is able to provide the piston with rotational motion [31,32]. As mentioned, bioprinting based on extrusion is known to be the most suitable, economical and typical approach because of its versatility and affordability. Therefore, there are several bioprinters commercially available on the market such as Tissue Scribe, BIOBOT$^{TM}$ BASIC, Engine HR, LulzBot Bio, Allevi, BIO V1, BIO X$^{TM}$ and many more [29,33] that rely on extrusion methods.

### 2.2. Droplet-Based Bioprinting

Compared to the extrusion technique, this type of bioprinting is based on the production of individual small droplets resulting in high-resolution 3D-printed structures. Furthermore, according to the formation method of the droplets, this type of bioprinting can

be separated into bioprinting based on inkjet, electrohydrodynamic jetting and bioprinting aided by laser [33].

The first one is the process of inkjet printing by the formation of individual droplets that are aimed at a defined location, and the subsequent droplet–substrate interaction. Furthermore, inkjet bioprinting may be divided into two modes: continuous and drop-on demand. Continuous inkjet relies on a phenomenon that displays the innate tendency of a stream of liquid to suffer a morphological transformation to become a line of individual drops. The ink provided in continuous inkjet is usually conductive and is driven by magnetic or electric forces [34]. Inkjet printing based on drop-on demand is classified into thermal, piezoelectric, and electrostatic, where such a division is made based on droplet motivation mechanisms [35,36]. This technology has enabled applications in industry ranging from microelectronics to the manufacturing of ceramics and biomedicine [37,38]. Another type of droplet bioprinting is thermal inkjet bioprinting where the thermal actuator is heated by a manageable impulsive voltage which leads to partial vaporization and formation of small bubbles [38]. Piezoelectric inkjet bioprinting, applies an actuator of piezoelectric nature in order to create droplets. In this case, the voltage provokes a rapid change in the chamber volume that results in the generation of acoustic waves, that provide a pulse of pressure for bioinking [39]. Electrostatic inkjet bioprinting is driven by the voltage given to a motor and a platen, provoking a bend on a platen that produces the bioink by extrusion. This technique displays an enhanced resolution of printing and a remarkable efficacy to eject inks of high viscosity which allows the bioprinting of, for example, gelatin. In this case, the broadness of the thinnest printed line was as small as 6 μm and was able to precisely print scaffolds for the culture of living cells [40].

Electrohydrodynamic is the second type of droplet-based bioprinting, where jetting is driven by filling a metallic nozzle with bioink for generating a meniscus of spherical shape at the nozzle tip. Next, a high voltage is produced between the substrate and the nozzle to generate an electrical field. As a result, droplets are expelled under a sufficient voltage when the electrostatic force breaks the surface tension [41].

Finally, laser-assisted bioprinting is a method that involves a contact-free and nozzle-free printing procedure to place biomaterials in a controlled manner onto the surface of materials. This type of printing includes LIFT, LGDW, AFA-LIFT, biological laser processing (BioLP), and matrix-assisted pulsed laser evaporation direct writing (MAPLE-DW) [42–44]. As an example, this technology has been used for the simultaneous bioprinting of mesenchymal stromal cells, together with collagen and nanohydroxyapatite, to improve the regeneration of bone tissue in a defect model in mice [43].

### 2.3. Photocuring-Based Bioprinting

This technique employs the photopolymerization properties of UV active polymers under accurately tailored UV radiation. Moreover, photocuring-based bioprinting includes two processes: stereolithography and digital light processing. Stereolithography printers have a tank that is filled with bioinks. Inside this tank, there is a flat structure that shifts up and down. Once the initial layer is printed, the platform moves to the superficial area of the solution of the bioink. Following, the liquid hardens point by point after exposition to the UV radiation. On the other side, digital light processing uses the same mechanism of stereolithography but solidifies a complete layer at once. Complex biostructures can be precisely printed with superb spatial exactitude, tailored physicochemical attributes and even with biochemically modified cells [45,46].

### 2.4. Biomaterials for 3D Bioprinting

With all the techniques specifically developed to print biomaterials, it becomes essential to match the best biomaterial to the most suitable technique in order to extract the best performance from the biomaterial. Moreover, it must be noticed that biomaterial-based bioink is one of the most critical components of 3D bioprinting due to its great impact on the generation of biological structures and especially on their behaviour. Within 3D

bioprinting materials, biocompatibility, homogeneous degradation, and easy printability are noteworthy aspects to consider. In general, biomaterials for 3D printing may be divided into natural and synthetic depending on their nature. Research on these materials proposed them as building blocks for 3D structures with the aim of repairing and replacing sections of organs of the body [47,48]. Within synthetic materials, there are the Polylactic Acid (PLA), Poly-D, L-Lactic Acid (PDLA), Acrylonitrile Butadiene Styrene (ABS), Polyethylene Glycol (PEG), Polyether Ether Ketone (PEEK), Polycaprolactone (PCL), PolyButylene Terephthalate (PBT), PolyUrethane (PU), PolyVinyl Alcohol (PVA), and PolyLactic-co-Glycolic Acid (PLGA) [49–51]. These synthetic polymers are identified for their mechanical properties to form defined structures due to their high strength.

On the other hand, natural biopolymers, such as alginate, gelatin, cellulose, hyaluronic acid, collagen or chitosan, are known for their high viscosity, biocompatibility, degradation, and low cost. These types of polymers have been used to develop bioinks, which are loaded with living cells, and biomolecules in a cellular-matrix environment [52,53].

Hence, the use of synthetic materials has spread in the last few years above the natural materials to solve the former problems of the last related to their scarce rheological properties, which are not optimal for printability. In this regard, bioinks must be nontoxic and printable at low temperatures since degradation of biomaterials might occur. Furthermore, it must be considered that using soft biomaterials based on water-friendly polymers can offer a cell-friendly environment for developing cell-laden structures and providing a conducive environment for cellular growth and development. Hydrogels with their high water content and tissue-mimicking properties have enabled their extended use in engineering of biological tissues [47,54]. In general, polymers capable of forming crosslinked structures such as hydrogels are a good selection due to their ability to retain an abundant quantity of water in them but keeping good mechanical properties. Thus, providing a physiological environment-type to biomolecules or cells.

## 3. Nanohydroxyapatite for Bioprinting

Fabrication of bone tissue substitutes can be performed using not only a single bioink but also a combination of bioinks in a composite material, combining the advantageous properties of each bioink in a synergic way to improve properties such as mechanical strength, printability, biocompatibility, and gelation characteristics. Hydroxyapatite (HA) has been widely investigated as a bioactive, osteoinductive and osteoconductive biomaterial, which makes it useful to be used as a porous replacement of damaged natural bone [55,56] (see Figure 2).

HA is a calcium phosphate. These inorganic salts have been thoroughly investigated and employed as inorganic fillers in bioprinting and the fact that around 60 wt% of bone is made of HA makes it an interesting material for research in this field [57]. Due to its chemical compositions and the release of specific ions when dissolved, HA can influence the differentiation of stem cells or progenitor cells leading to, biological responses, such as osteogenic responses. Furthermore, scaffolds for bone regeneration with accurately controlled sizes of pores and interconnectivity between them can be fabricated using 3D-printing techniques and HA-based bioinks.

Bone cements are composed of a self-curing acrylic polymer that consists of a powder fraction made up of the polymer, generally, methyl methacrylate, a polymerization initiator and a liquid fraction made up of the monomer. When the monomer and polymer are brought into contact, the monomer polymerization process begins. This process is progressive and the mixture, fluid at first, becomes pasty to obtain a resistant and non-reabsorbable solid material. The polymerization reaction is exothermic, reaching up to 80 °C, and is carried out in the operating room itself. This polymerization process is not optimal and therefore new technologies are demanded that can provide resorbable bone cements to be incorporated into the bone tissue. Bone cements can incorporate radiopaque chemical elements in their formulation with the function of revealing in the radiological examination the place where the cement was applied. Antibiotics may be added to bone

cement formulations so they function as prophylactics to reduce the incidence of infectious processes and as a treatment for prosthetic infections and other bone infections [58,59].

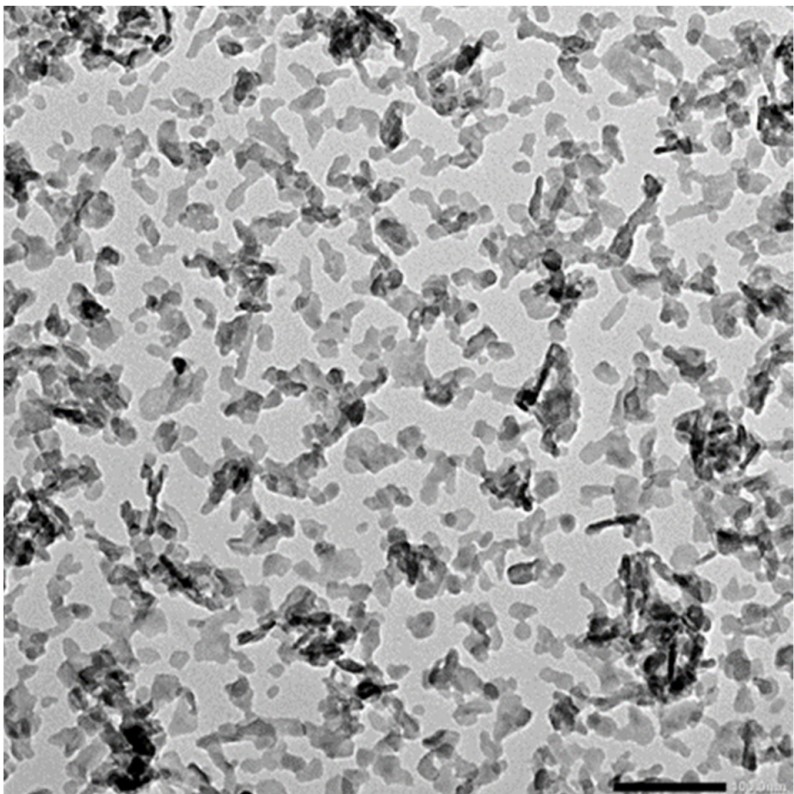

**Figure 2.** Transmission electron microscopy (TEM) image of synthetic hydroxyapatite nanoparticles.

Hydroxyapatite has appeared as an emerging bone cement material to promote bone formation and growth which has been demonstrated in various orthopaedic and dental applications [60]. HA displays advantages over other bone-filling materials such injectability, malleability and can be applied at body temperature. Furthermore, HA also performs controlled release of active substances [61,62] and may be used as an effective carrier of growth factors for osteoinduction, via sustained release [8]. HA is radiopaque which enables radiological examination [63] as well. Newly designed bone cements for bone tissue regeneration will attract new studies on HA-based bioinks, which are the base for bioprinted composite hydrogels incorporating HA particles. These composite bioinks, can provide in situ crosslinking effects and add for example mineralization capability as an extra functionality to the matrix. Intradermic injection of HA fillers has shown properties as collagen and elastin production, angiogenesis, and dermal cell proliferation [64]. Further functionalization with active molecules can enhance printability and local drug release properties for advanced therapies.

HA as single-component material for load-bearing applications gets limited usability due to its slow degradation and low mechanical strength. Nonetheless, HA particles can be used as reinforcing agents in tough and flexible polymer matrices [65], tailoring mechanical properties, which is of great interest for cells differentiation [66]. However, issues like specific ion release and its influence on cell viability, proliferation and migration have to be examined in the near future. In this regard, the interactions of the active filler components such as HA of 3D constructs and encapsulated cells by direct contact, on the overall (time-dependent) cell compatibility of the constructs have so far not been considered in detail in the published literature.

Although multi-scale porosity scaffolds are of special interest in 3D bioprinting there are several technical challenges encountered in developing 3D scaffolds of this kind, mainly

because of the challenge of making reproducible and controlled manufacture of a fully interconnected macro- micro-porous scaffold. One of the best approaches proposes the combination of porogen leaching and 3D printing, which results in the formation of microporosity in a controlled manner within the geometry of 3D-printed scaffolds [67]. The highly ordered 3D-printed macroporous scaffolds manufactured using this technique possess micropores whose dimensions are in the range of those beneficial for osteogenesis, as previously reported [68,69].

As described, the synergy between the HA chemical (composition and ions release) and morphological properties, enhances its bioactivity both in vitro and in vivo. Moreover, as mentioned earlier, hydroxyapatite is employed as a strategically incorporated ingredient to enhance the osteogenic properties of generally relatively inert polymeric biomaterials. Therefore, the manufacturing of a bioactive scaffold with HA, which displays nanoscale topographical features to enhance initial cell differentiation, microporosity for the accumulation of ions that promote biomineralisation, and a macro-porous network of polymer materials for facilitating vascularisation, represents a sound strategy for bone regeneration [70].

## 4. Conclusions

Recent literature on 3D printing of hydroxiapatite-based bioinks for various applications has been revised in this review. Three-dimensional bioprinting is an additive manufacturing procedure that uses biomaterials to build 3D objects with important applications in surgery and orthopaedics, reparation of internal organs and tissue engineering. Nowadays, bone repair and bone implants are the most advanced technological applications on the way to clinics.

Among the most popular printers, extrusion bioprinting is the most used technique in this field. Depending on the bioink, extrusion printers can fabricate scaffolds with defined shapes and controlled and interconnected porous structures. The fabrication of homogeneous material-based scaffolds in bone tissue engineering has involved the use of bioceramic–polymer composites and bioceramic–hydrogel mixtures. Biocompatible polymers from synthetic origin as PEG and PLGA and from natural sources have been broadly used as organic parts of composites and mixtures made of bioceramics and polymers for bone tissue engineering to fabricate elastic bones or artificial EC matrices to promote proliferation and cellular regeneration. Hydroxyapatite (HA) is the most widely used bioceramics due to its exceptional osteoconductivity and bioactivity as well as its bone resorption properties. The addition of HA to polymers and hydrogels improves the osteogenic and rheological properties of the printed composite platform enhancing mechanical consistency and form reliability. Moreover, the use of functionalized HA increases the bioavailability of certain ions of interest or other biological substances, through their sustained release over time.

Despite the high potential of this technique, 3D printing is still a very expensive and technically complex methodology. To continue advancing in the development of this new technology it is necessary to improve the optimization of the platform design, better knowledge of the cells and physiology of the organ and most importantly, the optimization of these hydroxyapatite-based biomaterials that can be printed and that can model the structural and functional complexity of human bone.

**Author Contributions:** Conceptualization, V.S.-M. and R.P.-G.; writing—original draft preparation, E.O., Á.G.-B., N.M.d.C. and V.S.-M.; writing—review and editing, V.S.-M., J.M.A., R.P.-G. and Á.G.-B.; funding acquisition, V.S.-M. and J.M.A. All authors have read and agreed to the published version of the manuscript.

**Funding:** This review was funded by i+Med S. Coop., Basque Government (HAZITEK program–HAPAZUR exp number ZL-2019/00548, ZL-2020/00461.

**Institutional Review Board Statement:** Not applicable.

**Informed Consent Statement:** Not applicable.

**Data Availability Statement:** The data presented in this study are available on request from the corresponding author.

**Conflicts of Interest:** All authors are employees of an company, I+Med S. Coop. In addition, Nagore Martínez de Cestafe, José María Alonso, Raúl Pérez-González and Virginia Sáez-Martínez own stocks in Company I+Med S. Coop.

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
