# Peer review of "Nanometric Hydroxyapatite Particles as Active Ingredient for Bioinks: A Review"

_2673-6209, doi:10.3390/macromol2010002_

Round 1
Reviewer 1 Report
The manuscript reviewed recent advances of nanometric hydroxyapatite particles as active ingredient for bioinks. The MS was well prepared.
Minor comments:
- the MS has many short paragraphs, I suggest to reorganize the paragraphs of the article;
- It is recommended to use some tables to summarize the research progress;
- In the references, 2021 publications are only 10%, I suggest to increase the 2020 or 2021 publications as references.
Author Response
Firstly, the authors would like to thank you for your time, dedication and professionality in reviewing this manuscript.
We really appreciate your comments and recommendations and thoroughly performed the corresponding modifications and corrections.
We have re-written several sentences and paragraphs to make the manuscript more easily readable.
In this regard, we have not included tables to summarize the information, but as explained, some of the new texts make the document more easy to follow.
Finally, we have increased the amount of references from years 2020-21.
Reviewer 2 Report
The paper is well written and easy to be understood also for people not expert in this particular field .Go on writing other papers on this relatively New subject
Author Response
Firstly, the authors would like to thank you for your time, dedication and professionality in reviewing this manuscript.
We really appreciate your comments and recommendations and thoroughly performed the corresponding modifications and corrections.
Thank you so much for your kind congratulations. We will of course, try to write more papers on this subject soon.